# *Acinetobacter nosocomialis* Causes as Severe Disease as *Acinetobacter baumannii* in Northeast Thailand: Underestimated Role of *A. nosocomialis* in Infection

Arnone Nithichanon,[a,b] Chidchamai Kewcharoenwong,[a,c] Hudadini Da-oh,[a] Sirithorn Surajinda,[a] Aranya Khongmee,[c] Surathinee Koosakunwat,[d] Brendan W. Wren,[e] Richard A. Stabler,[e] Jeremy S. Brown,[f] Ganjana Lertmemongkolchai[a,c]

[a]Department of Medical Technology, Faculty of Associated Medical Sciences, Chiang Mai University, Chiang Mai, Thailand
[b]Research and Diagnostic Center for Emerging Infectious Diseases (RCEID), Department of Microbiology, Faculty of Medicine, Khon Kaen University, Khon Kaen, Thailand
[c]Centre for Research and Development of Medical Diagnostic Laboratories, Faculty of Associated Medical Sciences, Khon Kaen University, Khon Kaen, Thailand
[d]Department of Medicine, Nakhon Phanom Hospital, Nakhon Phanom, Thailand
[e]Department of Infection Biology, London School of Hygiene and Tropical Medicine, Infectious and Tropical Disease, London, United Kingdom
[f]Centre for Inflammation and Tissue Repair, UCL Respiratory, London, United Kingdom

**ABSTRACT** Infections by *Acinetobacter* species are recognized as a serious global threat due to causing severe disease and their high levels of antibiotic resistance. *Acinetobacter baumannii* is the most prevalent pathogen in the genus, but infection by *Acinetobacter nosocomialis* has been reported widely. Diagnosis of patients with *A. baumannii* infection is often misdiagnosed with other *Acinetobacter* species, especially *A. nosocomialis*. This study investigated whether there were significant differences in clinical outcomes between patients infected with *A. baumannii* versus *A. nosocomialis* in Northeast Thailand, and to characterize serological responses to infection with these pathogens. The results show that *A. baumannii* had higher levels of multidrug resistance. Despite this, clinical outcomes for infection with *A. baumannii* or *A. nosocomialis* were similar with mortalities of 33% and 36%, respectively. Both pathogens caused community-acquired infections (*A. baumannii* 35% and *A. nosocomialis* 29% of cases). Plasma from uninfected healthy controls contained IgG antibody that recognized both organisms, and infected patients did not show a significantly enhanced antibody response from the first week versus 2 weeks later. Finally, the patterns of antigen recognition for plasma IgG were similar for patients infected with *A. baumannii* or *A. nosocomialis* infection, and distinct to the pattern for patients infected with non-*Acinetobacter*. In conclusion, our data revealed that infection with *A. nosocomialis* was associated with a similarly high level of mortality as infection with *A. baumannii*, the high rate of community-acquired infection and antibodies in uninfected individuals suggesting that there is significant community exposure to both pathogens.

**IMPORTANCE** Bacterial infections by *Acinetobacter* species are global threats due to their severity and high levels of antibiotic resistance. *A. baumannii* is the most common pathogen in the genus; however, infection by *A. nosocomialis* has also been widely reported but is thought to be less severe. In this study, we have prospectively investigated 48 reported cases of *A. baumannii* infection in Northeast Thailand, and characterized the serological responses to infection. We found that 14 (29%) of these infections were actually caused by *A. nosocomialis*. Furthermore, the incidence of antibiotic resistance among *A. nosocomialis* strains, APACHE II scores, and mortality for patients infected with *A. nosocomialis* were much higher than published data. Both *A. baumannii* and *A. nosocomialis* had unexpectedly mortality rates of over 30%, and both pathogens caused a high rate of community-acquired infections. Importantly, background antibodies in uninfected individuals suggest significant community exposure to both pathogens in the environment.

Address correspondence to Ganjana Lertmemongkolchai, ganjana.l@cmu.ac.th.

The authors declare no conflict of interest.

**KEYWORDS** *Acinetobacter baumannii*, *Acinetobacter nosocomialis*, multidrug resistance, anti-microbial resistance, community-acquired infection, clinical severity, antibody, antibody function

*A*cinetobacter species are Gram-negative bacteria, generally found in the soil and environment that can cause opportunistic infections in hospitalized and immunocompromised people (1, 2). The *Acinetobacter calcoaceticus-baumannii* (ACB) complex (3) is a cluster of pathogenic *Acinetobacter* species composed of *A. calcoaceticus*, *A. baumannii*, *A. nosocomialis*, *A. pittii*, *A. seifertii*, and *A. dijkshoorniae* (4–6). The most common clinical manifestations of bacterial infection by the ACB complex are pneumonia, sepsis, and skin and soft tissue infection. *A. baumannii* is the most well-known bacteria of the ACB complex due to high levels of antimicrobial resistance (AMR) and can cause a wide range of hospital acquired infections (7). Carbapenem-resistant *A. baumannii* was reported in 2016 as the most prevalent cause of hospital-acquired multidrug-resistant-infection due to Gram-negative bacteria in South East Asia (8). In Thailand, *A. baumannii* infection is associated with very high rates of carbapenem resistance of around 70% to 80%, and mortality rates more than 60% (8–10). As a consequence, *A. baumannii* infection is estimated to cause over 15,000 deaths per annum in Thailand (11). Due to the emergence of pan-antibiotic resistant *A. baumannii* and lack of alternative therapies, the World Health Organization (WHO) has identified *A. baumannii* as the most critically important bacteria that require improved prevention and therapeutic approaches as published https://www.who.int/news/item/27-02 -2017-who-publishes-list-of-bacteria-for-which-new-antibiotics-are-urgently-needed (12).

Infection with other *Acinetobacter* species is generally considered less severe compared with infection with *A. baumannii*, with lower levels of AMR and mortality. For example, patients with carbapenem-susceptible *A. baumannii* had higher 30-day mortality rate than infection with other *Acinetobacter* species (13), and infection with *A. baumannii* was also more frequently associated with admission to intensive care unit (14). These clinical data are supported by virulence studies using a *Galleria mellonella* infection model which have shown greater killing after infection with *A. baumannii* than *A. nosocomialis* or *A. pittii* (13). However, the data on the clinical differences in infection between species with the ACB complex are relatively limited, and may be confounded by misidentification of infecting *Acinetobacter* species using conventional microbiological techniques. Previous studies have reported that *A. pittii* and *A. nosocomialis* were identified in 24% to 66% of cases of ACB complex bacteremia, respectively (15).

Control of infection by the ACB complex is thought to depend on neutrophils, antibodies, and the activation of complement system (16). The ACB complex is not only a problem with multidrug resistance but also its survivability in the environment and evasion of human immunity (17). However, there are limited data on the dynamics of antibody development of people in endemic areas, or during infection with ACB. Experiments of mouse immunization with multiple strains of *A. baumannii* show a more diverse antibody profile with capacity to enhance bacterial clearance by neutrophil and improve defense against infection with several strains of *A. baumannii* (18). However, the human antibody profile has not been described.

In this study, we used a multiplex PCR to investigate whether *A. nosocomialis* infections are misreported as *A. baumannii* infection, compare the clinical significance of infection with *A. baumannii* or *A. nosocomialis*, and compare plasma IgG antibody responses in these patients.

## RESULTS

**Comparison of the clinical manifestations of infection with *A. nosocomialis* and *A. baumannii*.** Forty-eight patients with infections reported by the hospital laboratory as blood culture positive for *A. baumannii* infection were recruited from two hospitals in the Northeast of Thailand. In addition, 16 patients with non-*Acinetobacter* species infection, other bacterial infection, and 20 healthy controls were recruited at Srinagarind Hospital, Khon Kaen (Table 1). There were no differences between groups in age, gender,

**TABLE 1** General demographic data of participants

| Variable | Infected patients | | | Healthy controls (n = 20) | P value |
|---|---|---|---|---|---|
| | *A. baumannii* (n = 34) | *A. nosocomialis* (n = 14) | Other bacteria (n = 16) | | |
| Age, yrs, avg (SD)[a] | 60.4 (15) | 59.6 (22) | 62.9 (12) | 56.6 (11) | NS[b] |
| Female, *n* (%) | 15 (44) | 8 (57) | 7 (44) | 13 (65) | NS |
| Underlying condition, *n* (%) | | | | | |
| Diabetes | 11 (31) | 4 (29) | 9 (56) | 0 | NS |
| Thalassemia | 1 (2.8) | 1 (7) | 0 (0) | 0 | NS |
| Kidney disease | 5 (14) | 5 (36) | 9 (56) | 0 | 0.0053[h] |
| Hypertension | 11 (31) | 6 (43) | 9 (56) | 0 | NS |
| Reported organism culture positive, *n* (%) | AB,[d] 34 (100) | AB, 14 (100) | EC,[e] 8 (50) KP,[f] 3 (17) PA,[g] 3 (17) EC+KP, 2 (13) | NA | NA[c] |

[a]SD, standard deviation.
[b]NS, nonsignificant.
[c]NA, not applicable.
[d]AB, *A. baumannii*.
[e]EC, *E. coli*.
[f]KP, *K. pneumoniae*.
[g]PA, *P. aeruginosa*.
[h]Other bacteria groups compared to either *A. baumannii* or *A. nosocomialis* group.

and for the majority of underlying conditions with the exception of kidney disease, which was more common in patients with other bacterial infection compared with the *Acinetobacter* species infection groups (*P* value = 0.0053).

The causative bacterial isolate from each patient were analyzed using a multiplex PCR to confirm the infecting *Acinetobacter* species. The PCR did not identify *Acinetobacter* bands for *Klebsiella pneumoniae*, *Escherichia coli*, and *Pseudomonas aeruginosa* isolates (Fig. S3). Out of 48 *Acinetobacter* species isolates reported by the clinical laboratory as *A. baumannii*, multiplex PCR confirmed 34 (70.83%) as *A. baumannii* and the remaining 14 (29.17%) isolates were determined to be *A. nosocomialis*. The proportion of elderly patients and source of infection as not significantly different between patients infected with *A. baumannii* or *A. nosocomialis* (Table 2). The proportion of pulmonary, septicemic, and wound infections were 47%, 32%, and 29% for *A. baumannii*, and 36%, 7%, and 14% for *A. nosocomialis*, respectively. About one-third of patients had community-acquired infection. Patients with *A. baumannii* infection had higher odds ratios (OR) than *A. nosocomialis* infected patients for sepsis (OR 6.22), local wound infection (OR 2.50), and intensive care unit (ICU) admission (OR 2.09), although these differences did not reach statistical significance. Correlation analysis of sepsis compared between patients with *A. baumannii* and *A. nosocomialis* showed *P* values at 0.081 with two-tailed univariate analysis, and 0.069 with two-tailed multivariate correlation analysis. The severity score graded according to APACHE II criteria were comparable for patients with *A. baumannii* (score = 12.7) and *A. nosocomialis* (score = 12.4) infection. The mortality rates were also similar at 33.3% for *A. baumannii* and 35.7% for *A. nosocomialis*. Overall, these data suggest that isolation of *A. nosocomialis* from a blood culture was associated with a similar severity of illness as isolation of *A. baumannii*.

**Increased antimicrobial drug resistance for *A. baumannii* isolates compared with *A. nosocomialis*.** Hospital laboratory antimicrobial drug susceptibility data demonstrated that a high proportion of both the *A. baumannii* and *A. nosocomialis* isolates were resistant to antibiotics with 63% and 43% of isolates, respectively, described as MDR. *A. baumannii* isolates were more likely to be resistant to several antibiotics compared with the *A. nosocomialis* isolates, including amikacin (44% versus 7%, respectively, *P* value = 0.014), ciprofloxacin (59% versus 14%, respectively, *P* value = 0.005), piperacillin-tazobactam (65% versus 29%, respectively, *P* value = 0.036), and carbapenems resistance (61.7% versus 28.6%, *P* value = 0.036) (Table 3). Both MDR *A. baumannii*

**TABLE 2** A comparison of host factors and clinical characteristic between patients with *A. baumannii* and *A. nosocomialis* infection

| Variable | Infected patients | | Odd ratio (95% CI) | Univariate P value (two-tailed) | Multivariate P value (two-tailed) |
| | *A. baumannii* (n = 34) | *A. nosocomialis* (n = 14) | | | |
| --- | --- | --- | --- | --- | --- |
| Elderly age (*n*, %) | | | | | |
| ≧ 60 yrs | 17 (50) | 9 (64) | 0.56 (0.15 to 2.01) | 0.523 | 0.377 |
| < 60 yrs | 17 (50) | 5 (36) | 1.80 (0.50 to 6.50) | 0.526 | 0.377 |
| | | | | | |
| Source of infection (*n*, %) | | | | | |
| Hospital acquired | 22 (65) | 10 (71) | 0.73 (0.19 to 2.85) | 0.746 | 0.661 |
| Community acquired | 12 (35) | 4 (29) | 1.36 (0.35 to 5.30) | 0.746 | 0.661 |
| | | | | | |
| Clinical manifestation (*n*, %) | | | | | |
| Pulmonary | 16 (47) | 5 (36) | 1.60 (0.44 to 5.78) | 0.536 | 0.482 |
| Sepsis | 11 (32) | 1 (7) | 6.22 (0.72 to 53.79) | 0.081 | 0.069 |
| Skin wound | 10 (29) | 2 (14) | 2.50 (0.47 to 13.27) | 0.465 | 0.281 |
| | | | | | |
| ICU admission (*n*, %) | 23 (68) | 7 (50) | 2.09 (0.59 to 7.45) | 0.330 | 0.260 |
| Mortality rate (*n*, %) | 12 (33) | 5 (36) | 0.98 (0.27 to 3.60) | 1.000 | 0.978 |
| APACHE II score, mean (range) | 12.7 (2 to 31) | 12.4 (4 to 22) | ND[a] | 0.657 | 0.654 |

[a]ND, not determined.

and *A. nosocomialis* were associated with a higher mortality compared with non-MDR isolates (OR = 3.08 and 4.00, respectively) (Table 4).

**Recognition of *A. baumannii* and *A. nosocomialis* by plasma IgG antibody from infected patients.** The levels of plasma IgG to different isolates of *A. baumannii* and *A. nosocomialis* were quantified from infected patients using whole bacterial cell enzyme-linked immunosorbent assays (ELISAs). Initial assessment of plasma IgG from nine patients (at week 0 and week 2 after diagnosis) were performed against nine clinical isolates of *A. baumannii*, three clinical isolates of *A. nosocomialis*, and one standard isolate of *A. baumannii* ATCC 19606. Description of these clinical isolates are shown in Table S2. The degree of IgG binding to individual *Acinetobacter* strains varied between different patients. The results identified two patterns of IgG recognition for *A. baumannii*; patients with pattern 1 showed

**TABLE 3** Comparison of antimicrobial drug resistance profiles between *A. baumannii* and *A. nosocomialis* groups

| Variable | No. of resistant isolates (%) | | P value |
| | *A. baumannii* (n = 34) | *A. nosocomialis* (n = 14) | |
| --- | --- | --- | --- |
| Antimicrobial drug | | | |
| Amikacin | 15 (44) | 1 (7) | 0.014 |
| Ceftazidime | 21 (62) | 9 (64) | NS[a] |
| Ciprofloxacin | 20 (59) | 2 (14) | 0.005 |
| Gentamicin | 21 (62) | 8 (57) | NS |
| Carbapenems | 21 (62) | 4 (29) | 0.036 |
| Trimethoprim/sulfamethoxazole | 18 (53) | 9 (64) | NS |
| Piperacillin-tazobactam | 22 (65) | 4 (29) | 0.036 |
| Ceftriaxone | 23 (68) | 10 (71) | NS |
| | | | |
| Multidrug resistant (MDR) | 23 (68) | 6 (43) | NS |

[a]NS, nonsignificant.

**TABLE 4** Mortality rates of patients with *A. baumannii* (AB) or/and *A. nosocomialis* (AN) compared with multidrug resistance (MDR) and non-MDR isolates

| | No. of resistant isolates (%) | | | |
| --- | --- | --- | --- | --- |
| Variable | MDR-AB (*n* = 23) | Non-MDR-AB (*n* = 10) | *P* value[a] | Odds ratio (95% CI) |
| Mortality rate | 10 (43%) | 2 (20%) | 0.259 | 3.08 (0.60 to 16.39) |

| | No. of resistant isolates (%) | | | |
| --- | --- | --- | --- | --- |
| Variable | MDR-AN (*n* = 6) | Non-MDR-AN (*n* = 8) | *P* value | Odds ratio (95% CI) |
| Mortality rate | 3 (50%) | 2 (25%) | 0.580 | 4.00 (0.53 to 28.84) |

| | No. of resistant isolates (%) | | | |
| --- | --- | --- | --- | --- |
| Variable | MDR-AB+AN (*n* = 29) | Non-MDR-AB+AN (*n* = 8) | *P* value | Odds ratio (95% CI) |
| Mortality rate | 13 (44%) | 4 (22%) | 0.135 | 2.84 (0.82 to 9.24) |

[a]*P* value was analyzed with Fisher's exact test, while odds ratio was analyzed with Baptista-Pike method.

strong recognition of strains AB012, 015, 021, 028, and patients with pattern 2 showed strong recognition of strains AB001, 004, and 035. Patients showing strong recognition of *A. nosocomialis* strains belonged to the pattern 2 group for recognition of *A. baumannii* (Fig. S4). Therefore, for further evaluation of all patient's sera, we selected AB011, AB012, and AB035 clinical isolates and the ATCC 19606 strain as representatives of *A. baumannii* plus all three *A. nosocomialis* isolates. The heatmap of plasma IgG level from all samples showed that there was no clustering among participant groups, and there was a diverse pattern of antigen recognition for the three *A. baumannii* lysates (Fig. 1). The plasma IgG responses to the *A. nosocomialis* strains was more consistent (Fig. 1). In almost all participants, plasma IgG reacted against both *A. baumannii* and *A. nosocomialis* representatives (Fig. 1). The degree of recognition of the representative *Acinetobacter* strains varied between individuals, with *A. baumannii*-infected patients showing strong recognition of one strain that usually showed weaker recognition of the other *A. baumannii* strains. In contrast, *A. nosocomialis* patient sera showing high IgG recognition of *A. nosocomialis* tended to recognize all three strains investigated. A subset of *A. baumannii*-infected patients showed strong recognition of *A. nosocomialis* strains, and a subset of *A. nosocomialis*-infected patients showed strong recognition of *A. baumannii* strains. Interestingly, plasma IgG from some uninfected controls and patients infected with non-*Acinetobacter* species also showed high recognition of some *A. baumannii* and *A. nosocomialis* strains. To support the whole-cell ELISA data, immunoblots IgG antibody probing plasma IgG from infected patients were performed against the *A. baumannii* AB011, AB012, and AB035 strains and a representative *E. coli* strain. These demonstrated diverse patterns of recognition of strains AB011, AB012, and AB035 (Fig. 2). Small numbers of infected subjects did show similar patterns of antigen recognition against each of the strains (e.g., subjects AN026, AB022, AB035, and AB012 versus strain AB012), but overall there were no consistent differences between patients infected with *A. baumannii* or *A. nosocomialis*.

When measured by whole-cell ELISA, the strength of plasma IgG recognition of *A. baumannii* or *A. nosocomialis* strains generally did not show significant differences between *A. baumannii*, *A. nosocomialis*, and other pathogen infected groups, and to uninfected controls (Fig. S5). We investigated whether the plasma IgG level increased in response to *Acinetobacter* infection by comparing IgG ELISA data for samples obtained at diagnosis and after 2 weeks. The results demonstrated that the degree of plasma IgG binding and the pattern of binding to different strains from infected patients with *A. baumannii* or *A. nosocomialis* was similar for weeks 0 and 2 sera (Fig. 3). These data suggested that *A. baumannii* or *A. nosocomialis* infection did not enhance production of plasma IgG antibody against *Acinetobacter* species, and that instead there was significant pre-existing antibody to these *Acinetobacter* species.

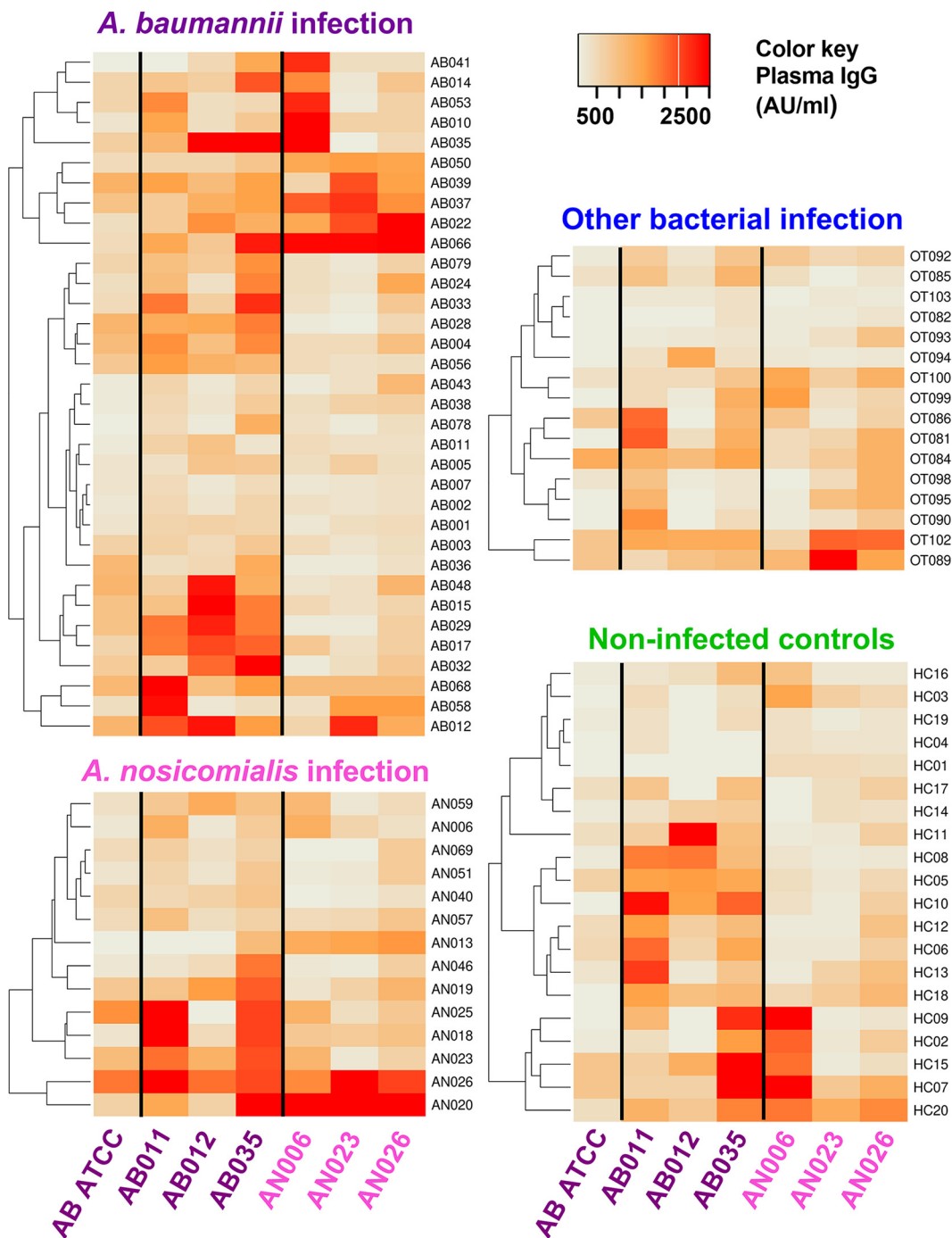

**FIG 1** Strength of recognition of selected *Acinetobacter* strain lysates by plasma IgG in sera from *A. baumannii*- and *A. nosocomialis*-infected patients and healthy controls measured by whole-cell ELISA. Paraformaldehyde fixed intact whole cells of *A. baumannii* (ATCC 19606, AB011, AB012, and AB035) or *A. nosocomialis* (AN006, AN023, and AN026) bacteria were coated onto plates at 10⁶ CFU. Plasma samples from acute infected patients with *A. baumannii* (n = 34; AB, purple), *A. nosocomialis* (n = 14; AN, pink), other bacteria (n = 16; OT, blue), or healthy controls (n = 20; HC, green) were added before detection of IgG binding by ELISA. Data of plasma IgG in AU/mL was visualized as heat maps with complete linkage clustering method and a distance measurement was completed using Euclidean method.

## DISCUSSION

The accurate identification of the causative infective agent is important for ensuring patients are treated appropriately. However, the diagnosis of patient with ACB complex infection using routine microbiology techniques is difficult (15), with previous reports suggesting misdiagnosis of non-*A. baumannii* species as *A. baumannii* occurred

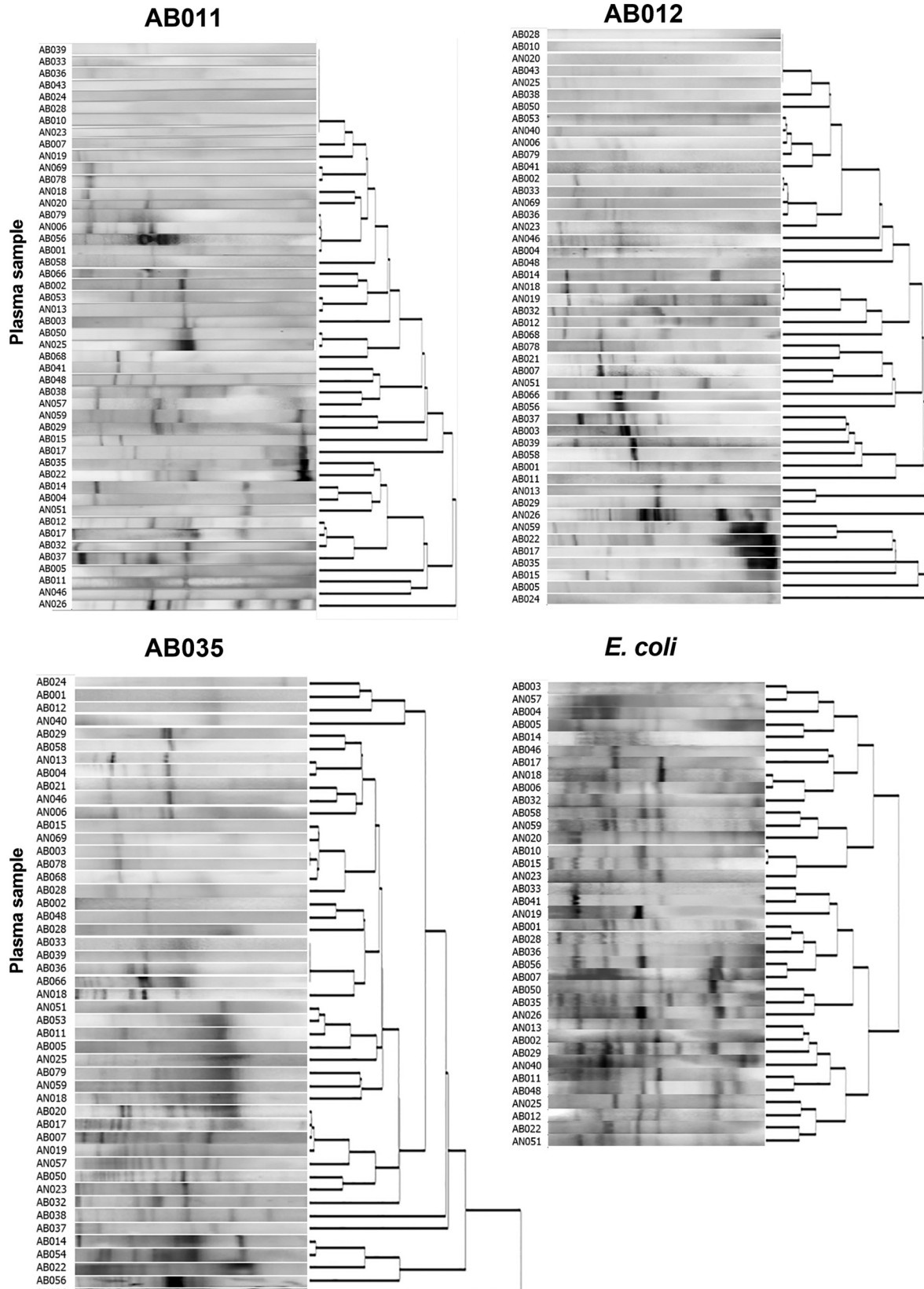

**FIG 2** Comparison of plasma IgG recognition patterns to lysate of *A. baumannii* or *E. coli* from patients with *Acinetobacter* infection. Whole-bacterial lysates of *A. baumannii* (AB011, AB012, or AB035) were separated by SDS-PAGE prior blotted onto nitrocellulose membrane and probed with plasma IgG from patients infected with *A. baumannii* (*n* = 36; AB patients) or *A. nosocomialis* (*n* = 16; AN patients). Representative clinical *E. coli* lysates were probed with plasma IgG from 24 *A. baumannii* patients and 13 *A. nosocomialis* patients.

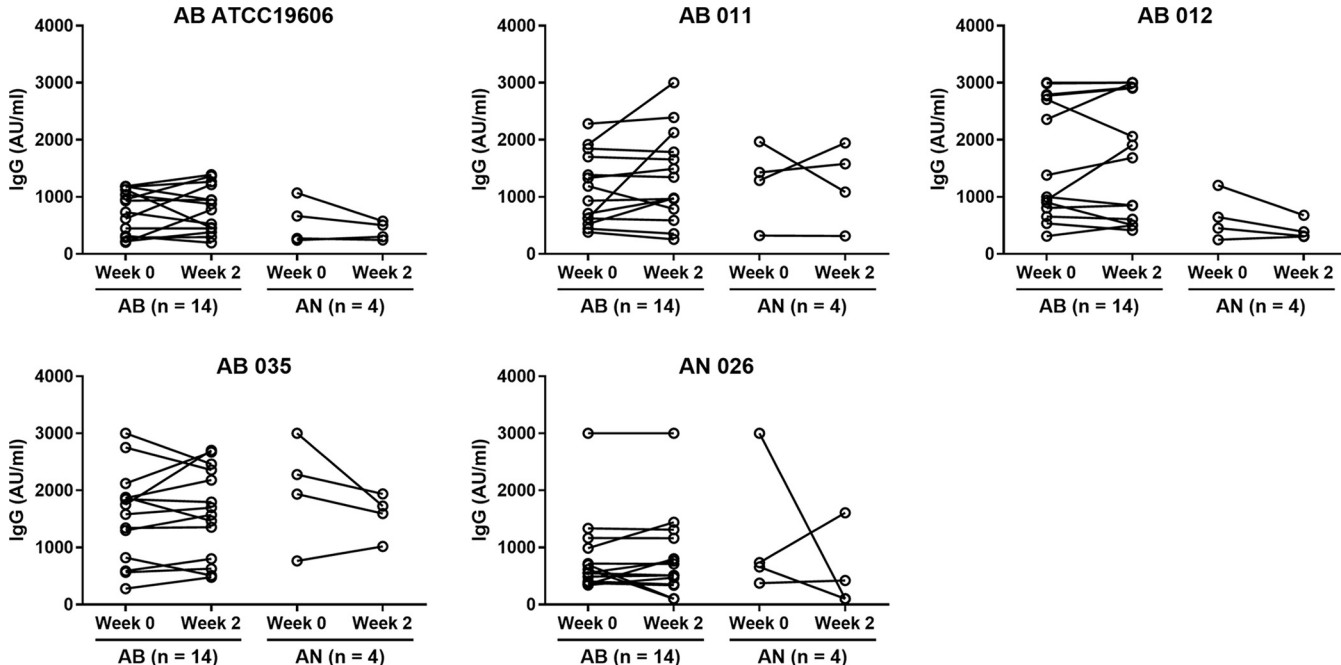

**FIG 3** Plasma IgG level against intact whole bacterial cells from patients with AB and AN infection at week 0 compared with week 2. Paraformaldehyde fixed intact whole cells of *A. baumannii* (AB ATCC, AB011, AB012, and AB035) or *A. nosocomialis* (AN026) bacteria were coated onto plate at $10^6$ CFU. Scatter dot plot and connection line with statistical test using Wilcoxon matched pairs signed rank test. ns, nonsignificant.

in 14% (18/111) and 11% (25/222) of cases in the United States (14) and Thailand (13), respectively. Here, using a multiplex PCR assay with species-specific primers identified by previous studies (19, 20) we show that in two Thai hospitals 29% of cases originally diagnosed by culture as *A. baumannii* were in fact *A. nosocomialis*. These data further demonstrate that misdiagnosis of ACB complex infection is common. Matrix-assisted laser desorption/ionization time-of-flight mass spectrometry (MALDI-TOF MS) could improve the accuracy of the diagnosis of ACB complex infections (21, 22), but is unavailable in most Thai hospitals. In this study, we also found that the *A. baumannii* isolates had higher rates of antimicrobial resistance than the *A. nosocomialis* isolates, specifically for amikacin, ciprofloxacin, piperacillin-tazobactam, and carbapenems. These results suggest that inaccurate microbiological diagnosis of *A. nosocomialis* as *A. baumannii* could lead to inappropriate treatment. However, both species still had very high rates of MDR at 68% for *A. baumannii* and 43% for *A. nosocomialis*. In contrast a previous study of intensive care patients in Northern China reported a larger differential in MDR rates between *A. baumannii* and *A. nosocomialis* at 86% and 21%, respectively (23), and in a previous Thai study all *A. nosocomialis* isolates were sensitive to carbapenems (13). The exact incidence of resistance to individual drugs among *A. nosocomialis* isolates is likely to vary if a larger number of isolates were analyzed compared with our data from a relatively small numbers of isolates. However, our data do demonstrate that MDR-*A. nosocomialis* isolates are not uncommon and this is an important observation when treating cases. The higher rates of antibiotic resistance for *A. nosocomialis* in our study demonstrate the importance of local resistance data when considering antibiotic treatment options and indicates that, similar to *A. baumannii*, *A. nosocomialis* also has significant potential to develop antibiotic resistance.

In further contrast to the previous Asian studies of ACB complex infections, our data demonstrated little difference in the severity of *A. nosocomialis* or *A. baumannii* infections. In our study, despite the higher incidence of antibiotic resistance and sepsis in *A. baumannii* infection compared with *A. nosocomialis*, the APACHE II illness severity score and mortality rate for patients infected with either organism were almost identical. The associated mortality of 36% for *A. nosocomialis* and 33% for *A. baumannii*

suggested infection with either organism was a serious clinical problem. The similarity of both APACHE II scores and the mortality would suggest even a larger case series from our centers would be unlikely to show significant differences in these parameters between patients with *A. nosocomialis* or *A. baumannii* infections. Previous data have previously shown a mortality for *A. baumannii* infections of 15% to 50%, markedly higher than 6% to 12% reported for *A. nosocomialis* infections (13, 23). Identifying whether the poorer outcomes of *A. nosocomialis* infections in our study are due to the increasing severity of *A. nosocomialis* infection over time or reflect a specific problem for our clinical sites will require further investigation, including comparative whole-genome sequencing studies of multiple *A. nosocomialis* strains and clinical data from other Asian centers.

Antibody facilitates clearance of *A. baumannii* through promoting phagocytosis and complement fixation (24–26), but there are very few data on the human antibody response to ACB complex bacteria. Our previous data for melioidosis, another endemic environmental bacterial infection in Thailand caused by *Burkholderia pseudomallei*, revealed a plasma IgG response to acute infection that was still detectable in the recovery stage compared with noninfected controls (27, 28). IgG responses to specific *B. pseudomallei* protein antigens were detected (29). Plasma antibody level correlated with the cellular immune responses as well as survival of melioidosis patients (27). Therefore, measuring plasma antibody can identify levels of protection against and/or assist diagnosis of infections with bacterial pathogens. We investigated plasma IgG antibody level against the whole cell of *Acinetobacter* bacteria and found markedly diverse plasma IgG responses in *A. baumannii*- and *A. nosocomialis*-infected patients against clinical *A. baumannii* and *A. nosocomialis* isolates. In contrast to other infectious diseases (30), our data showed the majority of our cases did not have a detectable change in plasma IgG to *A. baumannii* or *A. nosocomialis* isolates between the first week after a positive culture and 2 weeks later. Plasma IgG to representative *A. baumannii* isolates showed some grouping between clinical isolates, but high antibody levels to *A. baumannii* and *A. nosocomialis* were detected from some noninfected healthy controls. Overall, our results suggested that community exposure to *Acinetobacter* species could lead to pre-existing antibodies against *A. baumannii* and *A. nosocomialis*. This possibility was further supported by the significant rates of community-acquired *A. baumannii* and *A. nosocomialis* infection in our study, indicating that there is likely to be an environmental source of *Acinetobacter* infection in the locations used for our study which could also induce some degree of pre-existing immunity. Our data does not identify which antigens are recognized by plasma IgG, and it is possible that quantification of antibody to specific ACB complex proteins could identify markers that predict active infection or the clinical outcome of AB infection.

In conclusion, this study further demonstrated that conventional microbiology techniques cannot reliably discriminate *A. baumannii* from *A. nosocomialis*. In contrast to previous publications, both *A. nosocomialis* and *A. baumannii* infections had significant levels of antibiotic resistance and high levels of associated mortality. For the clinical management of patients with ACB infection, antimicrobial drug susceptibility testing remains a key to guide appropriate antibiotic therapy choices rather than identification of specific bacterial species. The high level of pre-existing plasma IgG antibodies seen in noninfected individuals and the large proportion of community-acquired infections suggest that people in the study areas are exposed to and can develop some adaptive immune responses against ACB complex pathogens. No effective vaccines are currently available for ACB complex pathogens, and our data provide both further evidence for the need for such a vaccine and demonstrates that significant adaptive responses to ACB complex pathogens do occur, which could be strengthened by vaccination.

## MATERIALS AND METHODS

**Ethics, participants, and data collection.** Procedures for sample and data collection were reviewed and approved by the Center for Ethics in Human Research, Khon Kaen University with approval number

HE611444 for participants collected at Srinagarind Hospital, Khon Kaen and by the Human Ethics Committee at Nakhon Phanom Hospital with the approval number NP-EC11-No.1/2562. Written informed consent was obtained from all participants and in compliance with the Declaration of Helsinki.

From March 11, 2019 to December 26, 2019, patients reporting a positive culture for *A. baumannii* were identified prospectively and enrolled (see demographic and clinical outcomes of patients in Table 1 and 2). Heparinized peripheral blood was collected within 24 h (week 0) and again on day 14 (week 2). The identity of the *Acinetobacter* species was confirmed using a multiplex PCR. Blood samples from patients who had positive bacterial cultures for non-*Acinetobacter* species were recruited as controls and classified as the "Other group." Healthy participants were defined by the guidelines for blood donation at the hospital and had no signs of infection at the time of blood collection.

Participant's medical history was collected, including age, sex, underlying conditions, ICU admission, and mortality (Table 1 and 2). Participants 60 years of age or over were classified as elderly age according to WHO policy (31). Patients with culture positive within 72 h of admission to hospital with no history of health care risks (dialysis, surgery, applying catheter) in the past 6 months were classified as community-acquired infection, whereas hospital-acquired infection was considered in patients with culture positive at greater than 72 h after admission (32). Patient disease severity was assessed by the international standard, Acute Physiology and Chronic Health Evaluation (APACHE II) score index assessed within 24 h before the positive culture of specimens (33). All clinical data were reviewed and reported by the senior nurse and confirmed by the clinical doctor at the hospital. Results of antimicrobial susceptibility testing were obtained. MDR-*A. baumannii* or -*A. nosocomialis* isolates were identified as those resistant to more than three classes of antibiotics (34).

**Bacterial preparation.** *Acinetobacter* isolates were grown in Luria-Bertani (LB) broth at 37°C to mid-log phase. After washing with phosphate buffer saline (PBS) pH 7.0, bacterial density was estimated using optical density measurement at 600 nm. In some experiments, bacteria were killed by incubation in 2% paraformaldehyde (PFA) for 60 min, washed, and kept frozen at −80°C until use. Bacterial lysates were prepared by incubating bacterial cultures in lysis buffer on ice for 20 min followed by centrifugation at $14,000 \times g$ for 5 min. Supernatants were collected and protein content measured using a NanoDrop spectrophotometer (Thermo Fisher Scientific). The lysates were stored at −80°C until use.

**\emph{A. baumannii}, \emph{A. nosocomialis}, or \emph{Acinetobacter} species identification by multiplex PCR and gel electrophoresis.** The *Acinetobacter* species PCR identification protocol was modified from previous studies (19). Briefly, a clinical isolate from each patient was separately grown on LB agar at 37°C overnight. Genomic DNA was extracted by using boiling method (35); approximately three to five isolate colonies were resuspended in sterile deionized water and then boiled at 95°C for 10 min. After centrifugation at $12,000 \times g$ for 10 min, supernatants were collected and estimated for DNA concentration by measuring the absorbance at 260 nm. PCR was performed according to the GoTaq Flexi DNA polymerase (Promega) manufacturers' instructions with GeneAmp PCR System 2700 (Applied Biosystems) thermocycle settings as follows: 94°C for 5 min, followed by 45 cycles of 94°C for 1 min, 60°C for 1 min, 72°C for 1 min, and a final extension at 72°C for 10 min. The primers used in this study and their interpretation are shown in Table S1.

The PCR amplicons were separated by electrophoresis (100 V, 80 min) in 1.5% agarose gel in 40 mM Tris, 20 mM boric acid, and 1 mM EDTA (TBE) buffer pH 8.3 containing DNA Gel Loading Dye (Thermo Fisher Scientific). The gels were visualized and captured using a gel image analysis system (UVitec, Cambridge, United Kingdom). The multiplex PCR results were validated for the first 18 isolates using 16S rRNA sequencing (Macrogen, Inc., South Korea) (Fig. S1).

**Plasma human IgG antibody quantification against whole intact cell of clinical *A. baumannii* and *A. nosocomialis* by ELISA.** The protocol for detection of plasma IgG against intact whole-cell bacteria was performed as previously reported (29, 36). Individual wells of 96-well polystyrene plates were coated with $10^6$ CFU of PFA killed bacteria in carbonate-bicarbonate buffer pH 9.6 overnight. After washing with 0.1% Tween 20 in PBS, the plate was blocked nonspecific binding with 10% fetal bovine serum (FBS) in PBS for 2 h at room temperature. Heparinized plasma was diluted 1:50 in 0.05% Tween 20, 10% FBS in PBS before adding to the plate, and incubating for 2 h at room temperature. The plate was washed and added for biotinylated goat anti-human IgG and HRP conjugated streptavidin (BD Biosciences) and then incubated 1 h at room temperature. After washing, color was developed for 15 min by using TMB Substrate Reagent Set (BD Biosciences). The reaction was then stopped by adding 2 N $H_2SO_4$. Absorbances were measured at 450 nm. The results were analyzed and shown as arbitrary units/mL (AU/mL) by comparing it with absorbances from an in-house prepared human IgG standard curve.

**Plasma human IgG antibody pattern profiling against lysate of clinical *A. baumannii* by Western blotting.** Bacterial lysates of clinical *Acinetobacter* species (500 $\mu$g) were separated by 12% to sodium dodecyl sulfate-polyacrylamide gel electrophoresis (SDS-PAGE). The proteins were then blotted onto a PVDF membrane with a wet electrophoresis system (Bio-Rad, Hercules, CA, USA), and the membrane was blocked with 5% skimmed milk for 1 h at room temperature. Heparinized plasma was diluted to 1:100 with 0.1% TBST, then added onto the membrane and incubated overnight. The membrane was washed with 0.1% TBST and biotinylated goat anti-human IgG and HRP conjugated streptavidin (BD Biosciences) was added for 1 h at room temperature. The membrane was washed and SuperSignal West Femto (Thermo Fisher Scientific) was added for signal detection, captured using a ChemiDoc XRS imaging system (Bio-Rad) and analyzed with Quantity One (Bio-Rad) software. Densitometry of individual bands was performed using the GelAnalyzer software. Coomassie blue staining of bacterial lysates and IgG binding on Western blot are shown in Fig. S2.

**Statistical data analysis.** All statistical analysis was done by using GraphPad Prism version 9 (GraphPad software). Contingency of category data were analyzed by chi-square, while differences in continuous data

were analyzed using two-tailed Mann-Whitney U tests. Multivariate correlation analysis was performed using two-tailed correlation matrix. Comparison of multiple groups was done using one-way ANOVA with Tukey's multiple-comparison test. Statistical significance was defined at $P$ value $< 0.05$. Heatmaps were generated using Heatmapper (http://www.heatmapper.ca/expression/) with complete linkage clustering method and Euclidean distance measurement method (37).

## SUPPLEMENTAL MATERIAL

Supplemental material is available online only.
**SUPPLEMENTAL FILE 1**, PDF file, 0.9 MB.

## ACKNOWLEDGMENTS

This work was supported by MRC DPFS MR/S004394/1 to R.A.S., B.W.W., G.L., and J.S.B. The funders had no role in study design, data collection and interpretation, or the decision to submit the work for publication. J.S.B. is partially supported by the Department of Health's National Institute Health and Care Research Biomedical Research Centre funding to University College London Hospitals.

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
