## [Reviewer comments · Microbiology Spectrum]

Microbiology Spectrum

***Acinetobacter nosocomialis* causes as severe disease as *Acinetobacter baumannii* in Northeast Thailand – underestimated role of *A. nosocomialis* in infection**

Arnone Nithichanon, Chidchamai Kewcharoenwong, Hudadini Da-oh, Sirithorn Surajinda, Aranya Khongmee, Surathinee Koosakunwat, Brendan Wren, Richard Stabler, Jeremy Brown, and Ganjana Lertmemongkolchai

Corresponding Author(s): Ganjana Lertmemongkolchai, Chiang Mai University

Review Timeline:

Submission Date:	July 24, 2022
Editorial Decision:	September 2, 2022
Revision Received:	September 19, 2022
Accepted:	September 21, 2022

Editor: Ayush Kumar

Reviewer(s): Disclosure of reviewer identity is with reference to reviewer comments included in decision letter(s). The following individuals involved in review of your submission have agreed to reveal their identity: Xian-Zhi Li (Reviewer #2)

Transaction Report:

DOI: <https://doi.org/10.1128/spectrum.02836-22>

September 2, 2022

Dr. Ganjana Lertmemongkolchai
Chiang Mai University
Department of Medical Technology
Faculty of Associated Medical Sciencee
Chiang Mai University
Chiang Mai 50120
Thailand

Re: Spectrum02836-22 (*Acinetobacter nosocomialis* causes as severe disease as *Acinetobacter baumannii* in Northeast Thailand - underestimated role of *Acinetobacter nosocomialis* in infection)

Dear Dr. Ganjana Lertmemongkolchai:

Thank you for submitting your manuscript to Microbiology Spectrum. I have now received comments on your manuscript. I myself had a careful read of the manuscript and agree with Reviewer #2 that your findings are of merit. However, before I can make a decision on the manuscript, I would like you to address the comments provided by the reviewers. When submitting the revised version of your paper, please provide (1) point-by-point responses to the issues raised by the reviewers as file type "Response to Reviewers," not in your cover letter, and (2) a PDF file that indicates the changes from the original submission (by highlighting or underlining the changes) as file type "Marked Up Manuscript - For Review Only". Please use this link to submit your revised manuscript - we strongly recommend that you submit your paper within the next 60 days or reach out to me. Detailed instructions on submitting your revised paper are below.

Link Not Available

Sincerely,

Ayush Kumar

Journals Department
Reviewer comments:

Reviewer #1 (Comments for the Author):

Arnone et.al investigated the role of *Acinetobacter nosocomialis* in infection. The mortalities resistance ratio and plasma IgG was evaluated. It is a interesting clinical study in *Acinetobacter*, but it is quite preliminary. Due to the limitation in the cases, it would be difficult to evaluate the meaning of the study. The comments were listed below:

1. The author used pcr to identify the bacteria. It is ok for identification. But if the author performed WGS, more information would be obtained, e.g. MLST, resistance gene, virulence gene, and the phylogenetic relationship of the bacteria.
2. Fig 1 move to supplement fig or remove it.
3. Table 2: only single factor stat was used. How about the multiple factor stat?
4. Watermark was observed in the end of the page.
5. The bacteria name in the references should be italic.

Reviewer #2 (Comments for the Author):

This ms reports the regional investigation of clinical outcomes and serological response in patients with two *Acinetobacter* species (*A. baumannii* vs. *A. nosocomialis*) with little differences demonstrated. The approaches taken are straightforward. Given the importance of *Acinetobacter* infections, the comparative results are considered helpful to the literature. Thus, the observations are publishable. However, there are some places requiring clarity. Comments including editorial revisions are given below.

P4-L16-17. The description of WHO's list is not correct. *A. baumannii* belongs to one of the most critically important bacteria. Ref 12 should be replaced by WHO's document (<https://www.who.int/news/item/27-02-2017-who-publishes-list-of-bacteria-for-which-new-antibiotics-are-urgently-needed>).

P17-L21. "vaccination" is described a measure in preventing ACB infection. As written, the description can be misleading as there seem currently no effective vaccines available for the purpose. A revision is needed for clarity.

Discussion: The ms should discuss the limitations of the study, for example, the numbers of isolates compared are still quite small, which could be contributing to the different observations in literature. With respect to drug susceptibility differences and similar clinical outcome (virulence), can the available genome data of two organisms provide an explanation?

P3-Importance: there is a lot of repetition of the Abstract contents. Please revise to shorten the contents.

P1-L1-2. Title; Not italicize "cause as severe disease as". The second "*Acinetobacter nosocomialis*" to be written as "*A. nosocomialis*".

P2-L9. Higher levels of multidrug resistance.

P4-L11/P5-L8. "multidrug" is noted. But in Page 2-L9, "multi-drug" is used. For consistence, suggest using "multidrug" for the ms (as often used in ASM journals). Check the ms.

P5-L15. "PCR" is noted. But in P6-L13, "polymerase chain reaction (PCR)" is seen. If needed, the full spelling needs to be in its first appearance. P7-L19. Heading, just use "PCR" without the repeated full spelling.

P6-L20. "ICU" is noted and needs to be introduced in P4-L23. Add "of" before "60".

P7-L1. "h" is noted. But in various early paces, "hours" is used (e.g., P6-L22/L24). P8-L20/L22 etc, "hrs" used. P9-L10/L13, etc. "hr" seen. Check for consistency.

P7-L12/L14/L23-24, etc. "min" versus "minutes": Use one and check the ms for consistence.

P7-L7. Full spelling of "PBS" is noted. No need to re-spell out "PBS" in P8-L19.

P12-L5. Heading, write "resistance", not "resistant".

P12-L7/P17-L16. Change "sensitivity" to "susceptibility". Check the ms for similar situations if needed.

P12-L13-14: Do not reintroduce full spelling of "MDR", which is first seen in P7-L4.

Staff Comments:

Preparing Revision Guidelines

Please return the manuscript within 60 days; if you cannot complete the modification within this time period, please contact me. If you do not wish to modify the manuscript and prefer to submit it to another journal, please notify me of your decision immediately so that the manuscript may be formally withdrawn from consideration by Microbiology Spectrum.

Response to Reviewers

Editor

Thank you for submitting your manuscript to Microbiology Spectrum. I have now received comments on your manuscript. I myself had a careful read of the manuscript and agree with Reviewer #2 that your findings are of merit. However, before I can make a decision on the manuscript, I would like you to address the comments provided by the reviewers.....

Authors: We deeply appreciate your positive assessment that the findings in this study are of merit. Below are the point-by-point responses to both reviewers' comments and suggestions. We believe that the manuscript has been significantly improved as a consequence.

Reviewer #1 (Comments for the Author):

Arnone et.al investigated the role of Acinetobacter nosocomialis in infection. The mortalities resistance ratio and plasma IgG was evaluated. It is a interesting clinical study in Acinetobacter, but it is quite preliminary. Due to the limitation in the cases, it would be difficult to evaluate the meaning of the study. The comments were listed below:

Authors: Thank you for your constructive comments. We agree that a study including more patients would have greater strength. However, large studies of *Acinetobacter* infections are very hard to complete; for example a 16 year study by Davis et al in Australia, only found 41 cases (Davis JS, et al. Chest 2014;146(4):1038-1045), and a study by Rodriguez-Bano et al. required data from 25 hospitals in Spain to identify 221 cases (Rodríguez-Baño J, et al. Infect Control Hosp Epidemiol. 2004;25(10):819-24). A Thai study did identify 222 cases over two years (Chusri S, et al. Antimicrob Agents Chemother. 2014;58(7):4172-9 which is ref 13 in our manuscript), but that study was retrospective and identified a total of 18 cases due to *A. nosocomialis*, not dissimilar to the 14 cases we have identified at two Thai hospital sites in our prospective study. The main clinical messages from our data are that *A. nosocomialis* infections are often misdiagnosed as *A. baumannii* infections, and that in contrast to previous data there was both a higher incidence of antimicrobial resistance and poor outcomes in our cases infected with *A. nosocomialis*. Although a larger case series could affect the actual percentage of *A. nosocomialis* strains that exhibit a significant level of antibiotic resistance, the general point that MDR *A. nosocomialis* strains are not uncommon (43% of isolates in our series) is unlikely to change. Furthermore, the almost identical mortality and APACHE II scores for patients infected with *A. nosocomialis* and *A. baumannii* in our series means that only a very large study could show statistically significant differences for these factors between these two pathogens for our hospital sites. Hence in our view, pausing publication to substantially increase the case numbers will not substantially alter the clinical data and does not warrant delaying dissemination of the clinical messages. We have now added sentences to the discussion on the limitations of our case numbers (Page 15, Line 20-24 and Page 16, Line 10-19).

1. The author used pcr to identify the bacteria. It is ok for identification. But if the author performed WGS, more information would be obtained, e.g. MLST, resistance gene, virulence gene, and the phylogenetic relationship of the bacteria.

Authors: We agree that WGS would provide more information on the resistance and virulence genes with phylogenetic relationship of the bacteria. We are experienced in WGS studies (Loraine J, et al. Front Microbiol 2020;11:548. doi: 10.3389/fmicb.2020.00548. eCollection 2020), and are planning WGS for the isolates reported in this manuscript. However, there are

substantial difficulties for ethical approval when performing WGS using patient information that will delay these data becoming available for an estimated 6-12 months. Furthermore, the strength in WGS is comparative data between different isolates which for our study would mean including a significant number of *A. nosocomialis* strains from outside of our clinical centers and at present these isolates are completely unavailable. This study investigates significant differences in clinical manifestations and outcomes between patients infected with *A. baumannii* versus *A. nosocomialis* and characterization of host immune responses to these bacteria. Overall, we feel that WGS data is somewhat tangential to the main messages of the paper and more suitable for a follow up publication exploring the interactions between host immune responses, clinical outcomes and genome data. We have inserted a sentence in the discussion relating to this (Page 16, Line 15-19).

2. *Fig 1 move to supplement fig or remove it.*

Authors: Figure 1 has been moved to Supplementary Figure S3, as suggested.

3. *Table 2: only single factor stat was used. How about the multiple factor stat?*

Authors: We performed a multivariate correlation analysis and inserted the details in the Methods (Statistical data analysis) as following: “Multivariate correlation analysis was performed using two-tailed correlation matrix. (Page 10, Line 1-2)”. The results were updated in the last column of Table 2 (Multivariate P-value (two-tailed)) and the following statement is in the Results: “Correlation analysis of sepsis compared between patients with *A. baumannii* and *A. nosocomialis* showed P values at 0.081 with two-tailed univariate analysis, and 0.069 with two-tailed multivariate correlation analysis.” (Page 11, Line 24 and Page 12, Line 1-2).

4. *Watermark was observed in the end of the page.*

Authors: The watermark was removed.

5. *The bacteria name in the references should be italic.*

Authors: The bacterial names in the references were corrected as italic.

Reviewer #2 (Comments for the Author):

*This ms reports the regional investigation of clinical outcomes and serological response in patients with two Acinetobacter species (*A. baumannii* vs. *A. nosocomialis*) with little differences demonstrated. The approaches taken are straightforward. Given the importance of Acinetobacter infections, the comparative results are considered helpful to the literature. Thus, the observations are publishable. However, there are some places requiring clarity. Comments including editorial revisions are given below.*

Authors: We welcome the reviewer’s positive comments and constructive suggestions. The manuscript has been revised accordingly. Below are the point-by-point responses.

*P4-L16-17. The description of WHO's list is not correct. *A. baumannii* belongs to one of the most critically important bacteria. Ref 12 should be replaced by WHO's document (<https://www.who.int/news/item/27-02-2017-who-publishes-list-of-bacteria-for-which-new-antibiotics-are-urgently-needed>).*

Authors: Thank you for providing the WHO precise weblink. We inserted the suggested link in the Introduction with a publication from WHO (Ref.12) according to the journal guidelines for

authors <https://journals.asm.org/journal/spectrum/article-types>. The inserted statement is as following: “as published <<https://www.who.int/news/item/27-02-2017-who-publishes-list-of-bacteria-for-which-new-antibiotics-are-urgently-needed>>” (Page 4, Line 17-19).

P17-L21. "vaccination" is described a measure in preventing ACB infection. As written, the description can be misleading as there seem currently no effective vaccines available for the purpose. A revision is needed for clarity.

Authors: To clarify this point, the following statement has been inserted: “No effective vaccines are currently available for ACB complex pathogens, and our data provide both further evidence for the need for such a vaccine and demonstrates that significant adaptive responses to ACB complex pathogens do occur which could be strengthened by vaccination.” (Page 18, Line 7-9).

Discussion: The ms should discuss the limitations of the study, for example, the numbers of isolates compared are still quite small, which could be contributing to the different observations in literature. With respect to drug susceptibility differences and similar clinical outcome (virulence), can the available genome data of two organisms provide an explanation?

Authors: We agree with the reviewer’s comments on numbers, and have addressed this point as described above in reply to a similar point from Reviewer #1. We also agree that WGS could identify whether the poor outcomes with *A. nosocomialis* in our case series are due to specific strains and have inserted a sentence into the discussion as stated above in reply to Reviewer #1’s comments.

P3-Importance: there is a lot of repetition of the Abstract contents. Please revise to shorten the contents.

Authors: The Importance has been rewritten, as suggested.

P1-L1-2. Title; Not italicize "cause as severe disease as". The second "Acinetobacter nosocomialis" to be written as "A. nosocomialis".

Authors: We corrected "cause as severe disease as" as indicated and changed the second "Acinetobacter nosocomialis" to "A. nosocomialis" as suggested (Page 1, Line 1-2).

P2-L9. Higher levels of multidrug resistance.

Authors: We corrected “multi-drug resistance” to “multidrug resistance” as suggested (Page 2, Line 9).

P4-L11/P5-L8. "multidrug" is noted. But in Page 2-L9, "multi-drug" is used. For consistence, suggest using "multidrug" for the ms (as often used in ASM journals). Check the ms.

Authors: We corrected “multi-drug” to “multidrug” throughout the manuscript.

P5-L15. "PCR" is noted. But in P6-L13, "polymerase chain reaction (PCR)" is seen. If needed, the full spelling needs to be in its first appearance. P7-L19. Heading, just use "PCR" without the repeated full spelling.

Authors: We added a full name of PCR at the first appearance (Page 5, Line 18) as following: “In this study we used a multiplex polymerase chain reaction (PCR)”. And we removed a full name from the heading (Page 7, Line 18) as “identification by multiplex PCR”.

P6-L20. "ICU" is noted and needs to be introduced in P4-L23. Add "of" before "60".

Authors: We added a full name of ICU at the first appearance (Page 6, Line 19), as “intensive care unit (ICU) admission”.

P7-L1. "h" is noted. But in various early paces, "hours" is used (e.g., P6-L22/L24). P8-L20/L22 etc, "hrs" used. P9-L10/L13, etc. "hr" seen. Check for consistency.

Authors: We corrected "h", "hrs" and "hr" as “h” throughout the manuscript.

P7-L12/L14/L23-24, etc. "min" versus "minutes": Use one and check the ms for consistence.

Authors: We corrected "minutes" as "min" throughout the manuscript.

P7-L7. Full spelling of "PBS" is noted. No need to re-spell out "PBS" in P8-L19.

Authors: We removed a full name of PBS at the second appearance, (Page 8, Line 20), “with 0.1% Tween-20 in PBS”.

P12-L5. Heading, write "resistance", not "resistant".

Authors: We corrected the heading, (Page 12, Line 9), as “Increased antimicrobial drug resistance for *A. baumannii* isolates”.

P12-L7/P17-L16. Change "sensitivity" to "susceptibility". Check the ms for similar situations if needed.

Authors: We changed "sensitivity" to "susceptibility", (Page 12, Line 11), as “Hospital laboratory antimicrobial drug susceptibility data” and (Page 18, Line 2), as “antimicrobial drug susceptibility testing remains a key to guide.....”

P12-L13-14: Do not reintroduce full spelling of "MDR", which is first seen in P7-L4.

Authors: We removed a full name of MDR, Page 12, Line 17, as “Both MDR *A. baumannii* and *A. nosocomialis*.....”.

September 21, 2022

Dr. Ganjana Lertmemongkolchai
Chiang Mai University
Department of Medical Technology
Faculty of Associated Medical Sciencee
Chiang Mai University
Chiang Mai 50120
Thailand

Re: Spectrum02836-22R1 (*Acinetobacter nosocomialis* causes as severe disease as *Acinetobacter baumannii* in Northeast Thailand - underestimated role of *A. nosocomialis* in infection)

Dear Dr. Ganjana Lertmemongkolchai:

Your manuscript has been accepted, and I am forwarding it to the ASM Journals Department for publication. You will be notified when your proofs are ready to be viewed.

Sincerely,

Ayush Kumar
Editor, Microbiology Spectrum
